# Allelopathic Effect of *Serphidium kaschgaricum* (Krasch.) Poljak. Volatiles on Selected Species

**DOI:** 10.3390/plants10030495

**Published:** 2021-03-05

**Authors:** Shixing Zhou, Toshmatov Zokir, Yu Mei, Lijing Lei, Kai Shi, Ting Zou, Chi Zhang, Hua Shao

**Affiliations:** 1State Key Laboratory of Desert and Oasis Ecology, Xinjiang Institute of Ecology and Geography, Chinese Academy of Sciences, Urumqi 830011, China; zhoushixing16@mails.ucas.ac.cn (S.Z.); zokir_06@mail.ru (T.Z.); meiyu@ms.xjb.ac.cn (Y.M.); shikai19@mails.ucas.ac.cn (K.S.); zouting@ms.xjb.ac.cn (T.Z.); 2Shandong Provincial Key Laboratory of Water and Soil Conservation and Environmental Protection, College of Resources and Environment, Linyi University, Linyi 276000, China; 3University of Chinese Academy of Sciences, Beijing 100049, China; 4Research Center for Ecology and Environment of Central Asia, Xinjiang Institute of Ecology and Geography, Chinese Academy of Sciences, Urumqi 830011, China; 5Chemistry and Environment Science School, Yili Normal University, Yining 835000, China; leilijing5521@hotmail.com

**Keywords:** allelopathy, phytotoxicity, volatile oil, VOCs, *Seriphidium kaschgaricum*

## Abstract

The chemical profile and allelopathic effect of the volatile organic compounds (VOCs) produced by a dominant shrub *Serphidium kaschgaricum* (Krasch.) Poljak. growing in northwestern China was investigated for the first time. *Serphidium kaschgaricu* was found to release volatile compounds into the surroundings to affect other plants’ growth, with its VOCs suppressing root elongation of *Amaranthus retroflexus* L. and *Poa annua* L. by 65.47% and 60.37% at 10 g/1.5 L treatment, respectively. Meanwhile, volatile oils produced by stems, leaves, flowers and flowering shoots exhibited phytotoxic activity against *A. retroflexus* and *P. annua*. At 0.5 mg/mL, stem, leaf and flower oils significantly reduced seedling growth of the receiver plants, and 1.5 mg/mL oils nearly completely prohibited seed germination of both species. GC/MS analysis revealed that among the total 37 identified compounds in the oils, 19 of them were common, with eucalyptol (43.00%, 36.66%, 19.52%, and 38.68% in stem, leaf, flower and flowering shoot oils, respectively) and camphor (21.55%, 24.91%, 21.64%, and 23.35%, respectively) consistently being the dominant constituents in all oils. Eucalyptol, camphor and their mixture exhibited much weaker phytotoxicity compared with the volatile oils, implying that less abundant compounds in the volatile oil might contribute significantly to the oils’ activity. Our results suggested that *S. kaschgaricum* was capable of synthesizing and releasing allelopathic volatile compounds into the surroundings to affect neighboring plants’ growth, which might improve its competitiveness thus facilitate the establishment of dominance.

## 1. Introduction

The genus *Seriphidium (Bess.)* Poljak. (Compositae) comprises approximately 130 species that mainly distributes in continental Asia, Europe, North America and North Africa [1]. This genus contains all life forms except trees: annual, biennial, perennial herbs, subshrubs and shrubs, some large in stature [2]. Approximately 31 species and three varieties have been found in China, with Xinjiang province being the center of distribution, where 26 species were discovered [3]. Among them, *Seriphidium kaschgaricum* (Krasch.) Poljak. is a subshrub herb that widely distributes in the arid and semiarid regions as a dominant species [3]. Several members of this genus are used in folk medicine as antihelminthics [4]. It is also well known for the relevant antimalarial artemisinin and other pharmacological and economic aspects [5,6,7]. The main secondary metabolites isolated from different species of the genus *Seriphidium* include triterpenoid, sesquiterpenes, coumarins, flavonoids, chromones, anthraquinones, phenolic derivatives, polyphenols and their glycosides, balchanins, costunolides, vulgarin, pyridine derivatives, ceramides, biphenyls and isoflavones, etc. [8]. *Seriphidium* plants are able to produce volatiles with strong aroma, which is suspected to serve as allelopathic agents to suppress neighboring plants and favor their own growth [9].

Allelopathy refers to any direct and indirect harmful or beneficial effect by one plant on another through the production of chemical compounds that release into the nearby environment [10]. Allelopathy is considered a possible mechanism to facilitate the wide spread of some species especially exotic invasive species by releasing allelochemicals such as alkaloids, flavonoids, phenolics, terpenoids, etc. [11,12,13]. Compositae plants can negatively affect other plants, especially some invasive plants such as *Ambrosia artemisiifolia* L. [14], *Eupatorium adenophorum* (Spr.) [15], *Ageratum conyzoides* (L.) L. [16], *Parthenium hysterophorus* L. [17], *Xanthium italicum* Morretti, and so on [13]. For instance, leaf extracts of *P. hysterophorus* exhibited high phytotoxic, cytotoxic and photocytotoxic activity on seed germination and seedling growth of garden cress and annual ryegrass [17]. Sesquiterpenoids such as xanthatin, 1α, 5α-epoxyxanthatin, 4-epiisxanthanol and 4-epixanthanol were isolated from *X. spinosum* that were phytotoxic against the tested plants *Amaranthus retroflexus* L. and *Poa annua* L. [11]. Although allelopathy was not significantly related to lifespan, life form or domestication of the interacting plants, it became more negative with increasing phylogenetic distance, indicating that allelopathy might contribute to coexistence of closely related species (i.e., convergence) or dominance of single species [18].

Volatile organic compounds (VOCs) are a group of important allelochemicals that are found to act not only in their gaseous state but also after deposition into the soil [19,20]. A number of plants are capable of synthesizing VOCs to play key roles in defense against pathogenic fungi, attracting seed-disperser and pollinators and herbivores, interplant signaling and allelopathic action [21,22,23]. Indeed, volatile oils have been found to serve as determinants of vegetation patterning or regulatory factor of community structure via allelopathy [24,25]. Puig et al. found that the volatiles of *Eucalyptus globulus* leaves inhibited growth of lettuce (*Lactuca sativa* L.) before and after incorporation into the soil by 29~34% [26]. In another study, the VOCs produced by *Atriplex cana* Ledeb. negatively affect seedling development of *A. retroflexus* and *P. annua*, and 80 g of fresh *A. cana* leaves and stems in a 1.5 L airtight container almost completely prohibited seed germination of test plants [27]. Souza-Alonso et al. reported that VOCs emitted from *Acacia longifolia* led to a significant decrease in the stem and root length of treated species *Lolium multiflorum* L., *Trifolium subterraneum* L., and *Sinapis alba* L. [28].

Like other *Artemisia* species, *S. kaschgaricum* possesses a distinctive aroma, implying the possible production of volatile compounds. *Artemisia* plants including *S. kaschgaricum* can be found thriving in the arid and semiarid regions, which are ecologically important in their habitats due to their excellent wind-breaking and sand-fixing function; in some cases, they are important constructive species and dominant species in the deserts [29]. To investigate whether *S. kaschgaricum* can release volatiles with allelopathic potential, we plan to: (i) analyze the phytochemical profile of *S. kaschgaricum* volatile oils; (ii) evaluate the phytotoxic effect of *S. kaschgaricum* volatile oil as well as their major constituents.

## 2. Results

### 2.1. Allelopathic Potential of VOCs

The allelopathic effect of VOCs released by *S. kaschgaricum* was investigated against *A. retroflexus* (dicot) and *P. anuua* L. (monocot). At the lowest treatment (2 g/1.5 L), VOCs released by *S. kaschgaricum* started to inhibit seedling growth significantly, reducing root elongation by 25.11% and 26.31% for *A. retroflexus* and *P. anuua*, respectively. VOCs at 5 g/1.5 L container suppressed radical elongation of *A. retroflexus* and *P. anuua* by 59.19% and 51.09%, meanwhile 10 g/1.5 L treatment resulted in reduction on root elongation of *A. retroflexus* and *P. anuua* by 65.47% and 60.37%, respectively. However, the effect of VOCs on seedling height was weaker than that of root elongation: at the highest concentration (10 g/1.5 L) applied, the VOCs suppressed seedling height of *A. retroflexus* and *P. anuua* by 60.37% and 29.96%, respectively. The dicot *A. retroflexus* was apparently more sensitive than the monocot *P. annua*, whose IC_50_ (the inhibitory concentration required for 50% inhibition) values were 4.679 g/1.5 L and 10.295 g/1.5 L for *A. retroflexus* root and shoot, and 5.666 g/1.5 L, 82.199 g/1.5 L for *P. annua* root and shoot, respectively (Figure 1). It was evident that *S. kaschgaricum* was capable of producing and releasing volatile compounds into the environment to affect other plants’ growth; we therefore went on to extract this plant’s volatile oil to investigate which chemicals were responsible active agents.

### 2.2. Composition of the Volatile Oils

The yield and chemical composition of the volatile oils extracted from different parts were compared in order to determine which part produces more active volatiles; the results were listed in Table 1. The yield of stem, leaf, flower and flowering shoot oil was 0.31%, 0.65%, 0.84% and 0.75% (*v*/*w*, volume/fresh weight), respectively, indicating that flowers contain more volatile compounds than stems and leaves. Eucalyptol and camphor were the most abundant constituents in all four oils, with the content of eucalyptol ranging from 19.52% to 43%, and camphor ranging from 21.55% to 24.91%; these two compounds occupied 64.55%, 61.57%, 41.16%, 62.03% of stem, leaf, flower and flowering shoot volatile oils, respectively. Oxygenated monoterpenes were overwhelmingly dominant, accounting for 85.35%, 81.46%, 60.58% and 77.89% of the stem, leaf, flower and flowering shoot volatile oils. Monoterpene hydrocarbons were also abundant in all four oils, representing 8.99%, 10.45%, 13.62% and 12.69% in stem, leaf, flower and flowering shoot volatile oils, respectively. The chemical composition of flower oil possessed the highest diversity, with 34 compounds being identified from this oil, and the constituents β-myrcene, cis-geraniol, trans-carveyl acetate, geranyl acetate, germacrene D, γ-elemene and globulol were discovered only in flower oil; meanwhile, (+)-trans-chrysanthenyl acetate and δ-elemene were unique in the flowering shoot oil, whereas pinocarvone was present only in stem oil, and α-terpinene was only found in leaf oil.

### 2.3. Phytotoxic Activity of the Volatile Oils and Their Major Constituents

Phytotoxic activity of the volatile oils (concentrations tested ranged from 0.2 to 5 mg/mL), as well as their major constituents was determined by comparing their plant regulatory effect on *A. retroflexus* and *P. annua*. Starting from the lowest concentration (0.2 mg/mL), all the volatile oils began to show significant inhibitory activity on root growth of receiver plants except for stem oil on *A. retroflexus* and flowering shoot on *P. annua*, which promoted root elongation of *A. retroflexus* slightly by 1.16%. At 0.5 mg/mL, stem, leaf, flower and flowering shoot oils significantly suppressed root growth of *A. retroflexus* by 58.78%, 55.78%, 70.06%, 12.28%, and *P. annua* by 47.92%, 59.29%, 31.64%, 26.97%, respectively. Furthermore, when the concentration reached 3 and 5 mg/mL, all the volatile oils basically completely inhibited root development of two receiver plants (Figure 2).

Shoot development of test species showed a similar pattern as root growth but to a lesser extent. At 0.5 mg/mL, stem, leaf, flower and flowering shoot oils significantly suppressed shoot growth of *P. annua* by 33.94%, 55.73%, 34.01% and 8.60%, and 46.83%, 52.15% and 49.05% for *A. retroflexus*, respectively; however, the effect of flowering shoot oil on *A. retroflexus* shoot was not significant. At 3 mg/mL, the volatile oils totally prohibited seedling development of receiver species except the leaf oil, reducing shoot elongation by 46.83% for *A. retroflexus*. In addition, when the concentration reached 5 mg/mL, all volatile oils completely inhibited seedling growth (Figure 3).

On the other hand, the major components eucalyptol and camphor as well as their mixture expressed inferior activity compared with *S. kaschgaricum* volatile oils. At the concentration of 0.5 mg/mL, the major compounds and their mixture started to inhibit root growth of two receiver plants significantly; eucalyptol, camphor and their mixture reduced root elongation by 18.82%, 23.92% and 23.92% for *A. retroflexus*, and 33.33%, 40.20%, 38.42% for *P. annua*, respectively. When the concentration reached 5 mg/mL, eucalyptol, chmphor and their mixture significantly reduced root elongation by 29.38%, 63.13% and 33.13% for *A. retroflexus*, and 51.90%, 72.30% and 58.31% for *P. annua*, respectively. Although the major constituents and their mixture consistently suppressed root and shoot development of both receiver species at 3 mg/mL, it is noteworthy to mention that even at the highest concentration applied (5 mg/mL), seedling growth was still not completely inhibited; the major components and their mixture significantly suppressed shoot growth by 22.20%, 29.48% and 18.28% for *A. retroflexus*, and 63.80%, 68.55% and 68.55% for *P. annua*, respectively (Figure 4 and Figure 5).

Our results showed that *S. kaschgaricum* volatile oils produced by different plant parts and their major constituents possessed significant phytotoxic activity against receiver plants, and their strength was compared by calculating their IC_50_ values (Table 2). In general, stem, leaf, and flower oils exhibited similar effect, although flower oil showed slightly stronger activity on root growth of *A. retroflexus* compared with other oils with an IC_50_ value of 1.735 mg/mL; meanwhile, flowering shoot oil always exerted the weakest activity, with an IC_50_ value of 2.186 mg/mL on root length of *A. retroflexus*. Leaf oil had the strongest activity on root elongation of the monocot plant *P. annua*, with the IC_50_ value of 1.593 mg/mL. Both the major constituents and their mixture resulted in inferior effect compared with all the oils; the IC_50_ values of eucalyptol were in general higher than camphor, and their mixture did not show significant synergistic effect. It is thus concluded that the volatile oil produced by different plant parts exhibited similar activity; however, the flowering shoot oil had the weakest phytotoxic activity; the major constituents did not show equivalent activity of the oils, implying that the presence of minor active constituents might contribute to the oils’ activity.

## 3. Discussion

The origin of the genus *Seriphidium* was not resolved; however, recent studies based on molecular evidence suggested that this previously separated subgenera still should be classified into the genus *Artemisia*, whose taxonomy has long been controversial due to the morphological complexities of its species [30]. Our study is the first report on the chemical composition and phytotoxic activity of *S. kaschgaricum* VOCs. The yield of stem, leaf, flower and flowering shoot oil was 0.31%, 0.65%, 0.84% and 0.75%, implying that flower produces more volatile compounds than stems and leaves. However, the strength of the oils’ phytotoxicity were comparable to each other, possibly due to the fact that their chemical profiles were quite similar. Previously, there have been reports on the chemical profile of *Seriphidium* species. Shao et al. found that α-thujone, β-thujone, eucalyptol and camphor were the most abundant constituents of *S. terrae-albae* (Krasch.) Poljakov volatile oil [9]. The primary dominating constituents of *S. brevifolium* (Wall. ex DC.) Ling & Y. R. Ling volatile oil were 2-bornanone, eucalyptol, α-thujone and β-thujone [31]. In another study, the major components of *S. herba-alba* (Asso) Soják oil were found to be camphor, α-thujone, cis-chrysanthenyl acetate, β-thujone, davanone, chrysanthenone and eucalyptol [32]. Gilani et al. evaluated the composition of *S. kurramense* (Qazilb.) Y. R. Ling volatile oil collected from Pakistan, and they found that α-thujone, β-thujone, 1, 8-cineole and camphor were abundant. Apparently, eucalyptol (also known as 1, 8-cineole) was a common constituent in *Seriphidium* volatile oils [33].

In *S. kaschgaricum* volatile oils, the majority of the constituents were oxygenated compounds such as eucalyptol and camphor. Oxygenated terpenes have been reported as more bioactive compared with nonoxygenated ones due to the reactivity of the hydroxyl group [34,35,36]. There have been several reports on the plant growth inhibitory activities of eucalyptol and camphor [37,38,39,40]. Martino et al. compared the antigerminative activity of 27 monoterpenes and found that eucalyptol suppressed radicle elongation of both radish and garden cress in a significant way at the lowest concentration tested (10^−6^ M) [41]. Eucalyptol was found to exert postemergence herbicidal activity against ryegrass and radish in a dose-dependent manner, suppression of seedling growth was significant at and above 0.1 mol/L on radish, meanwhile application of eucalyptol prohibited seedling growth of ryegrass when the concentration was above 0.1 mol/L, with root suppression first occurring at 0.0316 mol/L and shoot suppression at 0.1 mol/L [42]. In another study, Abraham et al. found that camphor and eucalyptol did not inhibit seed germination of maize (*Zea mays* L.) but reduced fresh and/or dry weight of roots at 5.0 mM and above [40]. In another study, camphor suppressed radicle and shoot growth of curly cress (*Lepidium sativum* L.) at 250 mg/m^3^, showing a nonlinear trend [43]. In this study, the strength of phytotoxicity exerted by eucalyptol, camphor and their mixture was much weaker than the volatile oils, indicating that some constituents in the oils that were not abundant might contribute to the phytotoxic effect, which needs further investigation in the future.

Production of VOCs is species specific. Viros et al. compared the emission of biogenic VOCs produced by 16 Mediterranean species and discovered clear differences between species, even when they belong to the same genus; besides, litter that possessed terpene storage structures and known to store high terpene concentration did not always release the highest terpene emission rates, meanwhile species that do not possess such structures release only non terpenic emissions [44]. Meanwhile, VOCs synthesized by plants may exist both in the air and in the soil matrix; Yosef Friedjung et al. evaluated the chemical profile of volatiles emitted by three desert plants including *A. sieberi* and *A. Judaica* and found that there was a high correlation between the major volatile compounds in the plant extracts and those found in the soil and air samples; for instance, 1,8-cineole, α-terpineol, trans-thujone were detected as abundant constituents of the volatile composition of *A. sieberi* plant foliage, in the soil in which *A. sieberi* is growing, and in the air surrounding the plant [45]. In another study, Barney et al. quantified the concentrations of volatile monoterpenes with allelopathic potential from the North American invasive perennial *A. vulgaris* and found that soil monoterpene concentrations were 74-fold higher inside (35 ng g^−1^) and 19-fold higher at the edge (9 ng g^−1^), compared to outside the *A. vulgaris* stand (0.48 ng g^−1^); combined with other results, it is suggested that monoterpenes produced by *A. vulgaris* had little direct effect in their volatile gaseous state but are accumulated in the soil matrix within and bordering the *A. vulgaris* stand so as to negatively impact neighboring species’ growth and foster its invasion success [46].

In many cases, the chemical profiles of VOCs and EOs of the same species are quite similar. For instance, Dragull et al. found that limonene (46%) was the major constituent of the headspace volatiles collected from fresh leaves of *Pistacia vera* “Kerman”, which was also dominant in the leaf EO (78~81%) [47]. In a study investigating the volatile compounds produced by *Salvia uliginos*, β-caryophyllene (12.91% for leaves and 25.64% for flowers), bicyclogermacrene (16.91% for leaves and 31.26% for flowers), δ-elemene (5.09 for leaves and 13.98% for flowers) and germacrene D (not detected in leaves and 8.65% in flowers) were the major ingredients of the HS-SPME profiles of leaves and flowers, which was quite similar with the chemical composition of the EO obtained from the aerial parts that was also rich in bicyclogermacrene (16.3%), germacrene D (14.81) and β-caryophyllene (8.57%); however, δ-elemene was not found in the EO, instead, another compound –spathulenol, was barely detected in the leaves by HS-SPME, was abundant in the EO (12.66%) [48]. Zhigzhitzhapova et al. compared the chemical composition of the EO extracted by hydrodistillation and VOCs detected by headspace extraction of *A. vulgaris* L., and results showed that 1, 8 cineole, camphor and α- and β- thujone were the main constituents in both of them [49]. In another study, an extensive analysis of the EO obtained from French coriander fruits showed the presence of linalool as the major component (72%), with an absolute concentration in the essential oil of 412 g/L; also, the area specific emission rate of linalool was determined at 125 μg m^−2^ h^−1^, indicating that linalool was the most abundant compound of coriander fruits volatiles [50]. However, depending on the plant species and detection methods, VOCs and EOs might differ in terms of their composition; for example, Najar et al. reported that pyranoid (10.3%) and β-caryophyllene (6.6%) were the major compounds of the VOCs emitted from leaves of *Sambucus nigra* L., whereas benzaldehyde (17.8%), α-bulnesene (16.6%) and tetracosane (11.5%) were abundant in the EO extracted from leaves [51].

In most circumstances, the allelopathic effect of allelochemicals are species selective and concentration dependent. Santonja et al. found VOCs produced by *Pinus halepensis* exerted more potent effect on *Linum strictum* than *Lactuca sativa*; meanwhile, strength of allelopathy was enhanced along with the increase of VOC concentration [52]. In another study, Ercoli et al. evaluated the allelopathic effect of the aqueous extracts of *Secale cereal*, *Brassica juncea* and *Vicia villosa* on *A. retroflexus*, *Chenopodium album* and *Polygonum aviculare*, and found that the strength of inhibitory effect differed significantly among test species, and the magnitude of reduction on shoot growth was always lower than on root growth, which responded in a dose-dependent manner; the aqueous extract of *S. cereale* significantly reduced the shoot length of *A. retroflexus* by 25% at the concentration of 1:10 whereas by 83% at 1:5 [53]. VOCs produced by Compositae species have been previously reported to be alleloapthic. Tang et al. tested the allelopathic activity of VOCs released by the exotic invasive weed *Xanthium sibiricum* and found that at 80 g fresh plant materials/1.5L, root growth of *A. retroflexus* and *P. annua* were suppressed by 49.1% and 69.6%, respectively [54]. VOCs released by 20 g of *Ageratina adenophora* leaf litter significantly affected seed germination and seedling growth of *Bidens biternata* in glass chambers (interior diameter 190, height 100 mm) [55].

Previously, there were reports on the phytotoxic/allelopathic potential of volatile oils produced by *Artemisia* species. Jiang et al. studied the chemical profile and phytotoxic activity of the essential oil extracted from *A. sieversiana*, which led to the identification of α-thujone (64.46%) and eucalyptol (10.15%) as the most abundant constituents [56]. Both the major oil components and the essential oil possessed significant phytotoxic effect against the receiver species *A. retroflexus*, *Medicago sativa*, *P. annua* and *Pennisetum alopecuroides*, with their IC_50_ values ranged from 1.55~6.21 mg/mL (α-thujone), 1.42~17.81 mg/mL (eucalyptol), 0.23~1.05 mg/mL (the mixture of α-thujone and eucalyptol) and 1.89~4.69 mg/mL (the essential oil). Due to the fact that the strength of the major constituents’ mixture was more potent than each individual compound, it was proposed that there might be a synergistic effect between these two compounds. *Artemisia tridentata* Nutt. releases a highly biologically active substance, methyl jasmonate (MeJA), a compound known to function as both a germination inhibitor and promoter in laboratory studies [57]. The volatile oil produced by *A. pedemontana* subsp. assoana inhibited root and leaf growth of *Lolium perenne* L. (30% growth inhibition respect to the control) and stimulated the growth of *Lactuca sativa* L. root (53% growth stimulation respect to the control) at 0.4 μg/μL [58]. Tsubo et al. found the VOCs released by *A. adamsii* promoted photosynthesis of *Stipa krylovii* with enhanced stomatal conductance, and *S. krylovii* grew faster and consumed more water when exposed to the VOCs even with water deficiency [59]. Researchers found that *A. herba-alba* volatile chemicals inhibited seed germination of *Pinus halepensis* Mill. and reduced the root biomass of *Salsola vermiculata* L. seedlings with 0.5 g aerial parts of *A. pedemontana* in 10-cm-diameter Petri dishes, which coexist with *A. herba-alba* in natural semiarid plant communities [60]. Another study indicated that the negative allelopathic effect of *A. halodendronon* Turcz. ex Besser was responsible of triggering species replacement during the dune stabilization process in the Horqin Sandy Land [61]. Regarding the allelopathic potential of VOCs emitted by *Seriphidium* species, the volatile oil produced by *S. terrae-albae* (Krasch.) Poljakov and *S. kurramense* have been studied [9,62]. The volatile oil of *S. terrae-albae* reduced root and shoot length of *A. retroflexus* and *P. annua* by 99.35%, 99.5%, and 98.47%, 98.49% of the control at the concentration of 1.5 μL/mL, respectively [9]. In another study, volatile oils released by *S. kurramense* markedly affected the radical and hypocotyle growth of *Lemna minor* L. ranging from 87.1~100% and 84.3~100%, respectively [62].These studies suggested that Compositae plants including *Artemisia* species such as *S. kaschgaricum* might enhance its competitiveness by synthesizing bioactive VOCs with allelopathic potential into the surroundings to favor their dominance in the habitat.

## 4. Materials and Methods

### 4.1. Plant Material

Aboveground plant parts of *S. kaschgaricum* at the flowering stage were collected in suburban Urumqi, Xinjiang province, China in July 2017 (Lat 43.6546 N, Lon 87.3053 E, with an elevation of 627.57 m). Plants were identified by Prof. Yin Linke from Xinjiang Institute of Ecology and Geography, Chinese Academy of Sciences. Voucher specimens were deposited with the number of XJBI017075 at the herbarium of Xinjiang Institute of Ecology and Geography, Chinese Academy of Sciences. The plant material was separated into stems, leaves and flowers. Flowering shoots as well as plant parts were cut into small pieces and weighed before volatile oil extraction. Fresh plant materials were used directly for extracting volatile oils.

### 4.2. Potential Allelopathic Effect of VOCs

Fresh aerial plant parts of *S. kaschgaricum* were weighed and arranged at the bottom of airtight plastic containers (13.5 cm × 13.5 cm × 8.5 cm, volume 1.5 L) at the following ratios: 0 g, 2 g, 5 g, and 10 g per container according to published literature [54]. *Amaranthus retroflexus* L. and *Poa annua* L., a dicot and a monocot species that can be found growing in the same habitats with *S. kaschgaricum*, were chosen as the receiver species. These 2 species have frequently been adopted as receiver plants in allelopathic research because of their uniform seedling emergence, high germination rate and their significance in both agricultural and natural fields [9,27,37,53]. Their seeds were first surface sterilized using 0.5% HgCl_2_ and then sowed into Petri dishes (9 cm in diameter) lined with Whatman filter papers. Each Petri dish received 5 mL distilled water and were then transferred into the plastic containers. The containers were stored in a phytotron at 25 ± 2 °C with a photoperiod L/D = 16:8, which were kept open for 15 min each day to allow in fresh air for the seedlings. Seedlings were measured after 5 days (*A. retroflexus*) or 7 days (*P. annua*) incubation with 5 replicates (*n* = 50) [27,63].

### 4.3. Isolation of the Volatile Oil

Fresh aboveground plant parts were first separated into stems, leaves, flowers and then hydrodistillated separately to yield the stem, leaf, flower and flowering shoot volatile oils. For each organ, two hundred g of fresh materials were hydrodistillated for 4 h using a Clevenger type apparatus to extract the EO, and this procedure was repeated 3 times (altogether 600 g plant materials used for each plant organ) to yield enough oil for GC/MS analysis and the following bioassay, which were then dried over anhydrous sodium sulfate and kept at 4 °C until required [37].

### 4.4. Gas Chromatography-Mass Spectrometry Analysis

The oils were analyzed by GC/MS using a Perkin-Elemer Autosystem XL-Turbemass system with a PE-5MS capillary column (60 m × 0.25 mm; 0.25 μm film thickness) as previously described [45]. Samples were diluted in hexane before injection. Relative amounts of individual compounds were calculated based on their GC peak areas without FID response factor correction. Identification of the constituents of the volatile oil was achieved by comparison of their mass spectra and retention indices (RI, calculated by linear interpolation relative to retention times of a standard mixture of C_7_-C_40_
*n*-alkanes) with NIST (The National Institute of Standards and Technology) and a home-made library and those given in the literature [64,65].

### 4.5. Phytotoxic Effect of the Volatile Oils

Phytotoxicity of the volatile oils as well as the major constituents, i.e., eucalyptol, camphor and their mixture (eucalyptol: camphor = 34.47: 22.86, the average percentage of eucalyptol and camphor in the oils), were tested against *A. retroflexus* and *P. annua*. Eucalyptol and camphor (purity 98%) were purchased from Sigma-aldrich Co. (St. Louis, USA). The volatile oils, eucalyptol, camphor and their mixture were diluted in 0.5% acetone in distilled H_2_O to yield 0.2, 0.5, 1.5, 3, 5 mg/mL solutions, with acetone as the initial solvent (our previous experiment showed acetone at such concentration did not significantly affect seedling growth of the test plants [37,56]. Five ml of 0.5% acetone in distilled H_2_O (control) or diluted solutions were added to each petri dish (9 cm in diameter) which contained 10 test seeds. Petri dishes were stored in an incubator in the dark at 25 °C ± 2 °C. Seedlings were measured after 5 days for *A. retroflexus* and 7 days for *P. annua*. Five replicates were made for all phytotoxic bioassays. After this period, root and shoot length was measured (in total 50 seedlings) and the average growth in response to each treatment was calculated (expressed as a percentage of the average growth registered for the control sample). These values were used to calculate the IC_50_ (inhibition concentrations that cause a 50% growth registered for the control sample).

### 4.6. Statistical Analyses

The bioassay experiment followed a completely randomized design with five replications and 50 seedlings for each treatment. Results were expressed as mean ± standard error (SE) of the mean. One-way ANOVA (*p* < 0.05) was applied using the SPSS statistical package version 13.0 (SPSS Inc., Chicago, Illinois, USA) for Windows to examine whether the difference of the allelopathic/phytotoxic effects of the volatile oils produced by stems, leaves, flowers and flowering shoots, their major constituents, i.e., eucalyptol and camphor as well as their mixture tested at different concentrations was significant; then all the above-mentioned data was further processed using Fisher’s LSD test at *p* < 0.05 level to compare the difference among treatments. The inhibitory concentration required for 50% inhibition (IC_50_) values was calculated using probit analysis (SAS Institute. SAS/STAT User’s Guide).

## 5. Conclusions

Our results suggested that *S. kaschgaricum* was capable of synthesizing and releasing volatile compounds with allelopathic potential into the surroundings to affect other plants’ growth thus improve its competitiveness and facilitate the establishment of the dominance of this plant. Eucalyptol and camphor were the major constituents of the volatile oils; however, the strength of their phytotoxicity was much weaker than the volatile oils, implying that less abundant compounds in the volatile oil might contribute significantly to the oils’ activity.

## Figures and Tables

**Figure 1 plants-10-00495-f001:**
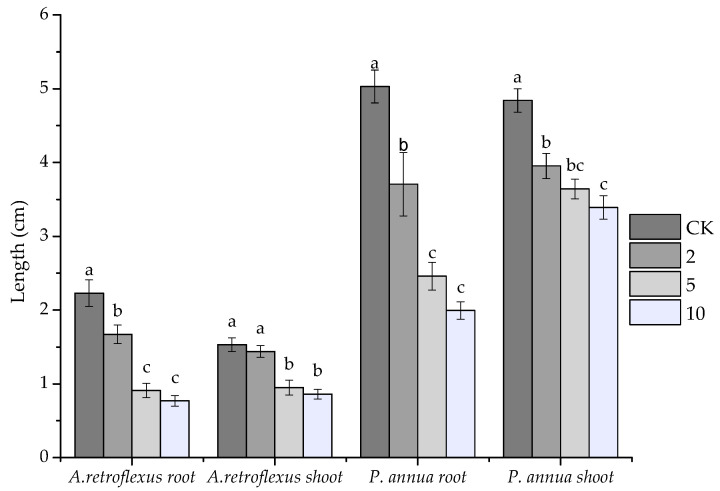
Allelopathic effect of volatile organic compounds (VOCs) released by fresh aerial parts of *S. kaschgaricum* tested at 0 g/1.5 L (CK), 2 g/1.5 L, 5 g/1.5 L and 10 g/1.5 L on root and shoot elongation of *A. retroflexus* and *P. annua*. Each value is the mean of five replicates ± SE (*n* = 50). Means with different letters (a, b, c, etc.) indicate significant differences at *p* < 0.05 level according to Fisher’s LSD test.

**Figure 2 plants-10-00495-f002:**
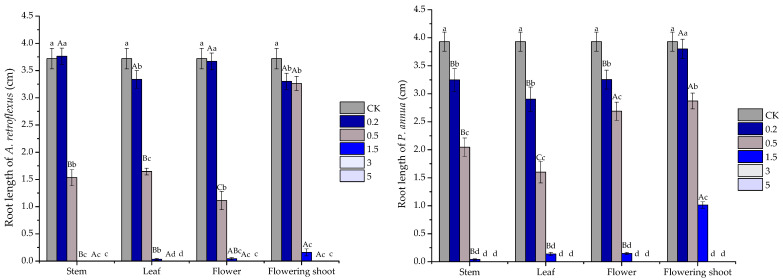
Phytotoxic effect of stem, leaf, flower and flowering shoot volatile oils of *S. kaschgaricum* on root elongation of *A. retroflexus* and *P. annua*. Each value is the mean of five replicates ± SE (*n* = 50). Different lowercase letters (a, b, c, etc.) indicate significant differences among different concentrations (0 mg/mL(CK), 0.2 mg/mL, 0.5 mg/mL, 1.5 mg/mL, 3 mg/mL and 5 mg/mL) of the same oil treatments at *p* < 0.05 level according to Fisher’s LSD test, and different uppercase letters (A, B, C, etc.) indicate significant differences among stem, leaf, flower and flowering shoot volatile oils at the same concentration (Fisher’s LSD test, *n* = 50, *p* < 0.05).

**Figure 3 plants-10-00495-f003:**
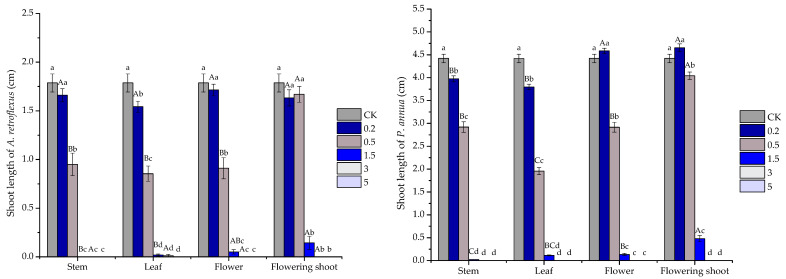
Phytotoxic effect of stem, leaf, flower and flowering shoot volatile oils of *S. kaschgaricum* on shoot development of *A. retroflexus* and *P. annua*. Each value is the mean of five replicates ± SE (*n* = 50). Different lowercase letters (a, b, c, etc.) indicate significant differences among different concentrations (0 mg/mL(CK), 0.2 mg/mL, 0.5 mg/mL, 1.5 mg/mL, 3 mg/mL and 5 mg/mL) of the same oil treatments at *p* < 0.05 level according to Fisher’s LSD test, and different uppercase letters (A, B, C, etc.) indicate significant differences among stem, leaf, flower and flowering shoot volatile oils at the same concentration (Fisher’s LSD test, *n* = 50, *p* < 0.05).

**Figure 4 plants-10-00495-f004:**
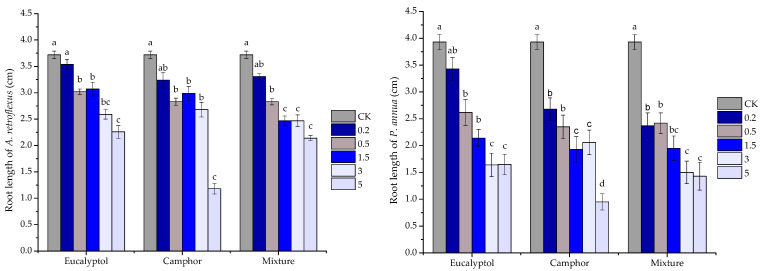
Phytotoxic effect of eucalyptol, camphor and their mixture at different concentrations (0 mg/mL(CK), 0.2 mg/mL, 0.5 mg/mL, 1.5 mg/mL, 3 mg/mL and 5 mg/mL) on root elongation of *A. retroflexus* and *P. annua*. Each value is the mean of five replicates ± SE (*n* = 50). Means with different letters (a, b, c, etc.) indicate significant differences at *p* < 0.05 level according to Fisher’s LSD test.

**Figure 5 plants-10-00495-f005:**
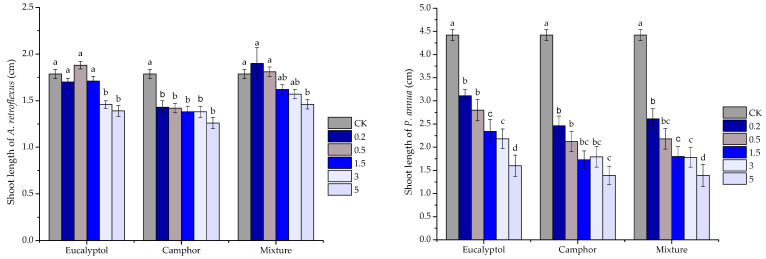
Phytotoxic effect of eucalyptol, camphor and their mixture at different concentrations (0 mg/mL(CK), 0.2 mg/mL, 0.5 mg/mL, 1.5 mg/mL, 3 mg/mL and 5 mg/mL) on shoot development of *A. retroflexus* and *P. annua*. Each value is the mean of five replicates ± SE (*n* = 50). Means with different letters (a, b, c, etc.) indicate significant differences at *p* < 0.05 level according to Fisher’s LSD test.

**Table 1 plants-10-00495-t001:** Chemical composition of the volatile oils.

Number	Compounds	Percentage (%)				Average Percentage (%)
		Stem	Leaf	Flower	Flowering Shoot	
1	Santolina triene	0.89	1.88	3.84	2.17	2.20
2	α-Pinene	1.86	1.43	1.91	1.6	1.70
3	Camphene	3.17	3.89	4.36	4.95	4.09
4	β-Pinene	0.78	0.9	1.17	0.92	0.94
5	Artemiseole	1.28	1.92	1.78	1.47	1.61
6	β-Myrcene	-	-	0.23	-	0.06
7	2-Carene	0.45	-	0.45	0.59	0.37
8	α-Terpinene	-	0.43	-	-	0.11
9	o-Cymene	1.02	1.18	0.59	1.35	1.04
10	Eucalyptol	43	36.66	19.52	38.68	34.47
11	γ-Terpinene	0.82	0.74	0.83	0.87	0.82
12	Terpinolene	-	-	0.24	0.24	0.12
13	Linalool	0.39	0.52	0.66	0.61	0.55
14	cis-2-p-Menthen-1-ol	0.43	0.34	0.23	-	0.25
15	Camphor	21.55	24.91	21.64	23.35	22.86
16	Nerol oxide	1.9	1.67	1.43	1.79	1.70
17	Pinocarvone	0.29	-	-	-	0.07
18	borneol	5.92	3.5	2.02	4.06	3.88
19	Lavandulol	0.23	0.31	0.4	0.27	0.30
20	2-Caren-4-ol	3.44	3.16	2.58	2.85	3.01
21	α-Terpineol	2.13	2.83	2.64	1.72	2.33
22	cis-Carveol	0.27	0.25	0.27	-	0.20
23	Nerol	2.11	2.82	3.41	1.36	2.43
24	Carvone	2.41	2.57	3.72	1.73	2.61
25	(+)-trans-Chrysanthenyl acetate	-	-	-	0.36	0.09
26	cis-Geraniol	-	-	0.28	-	0.07
27	Bornyl acetate	0.49	0.79	1.63	0.76	0.92
28	Lavandulol acetate	-	-	1.09	0.34	0.36
29	α-Terpineol acetate	2.46	1.32	1.87	-	1.41
30	δ-Elemene	-	-	-	0.96	0.24
31	Eugenol	0.23	0.4	0.47	-	0.28
32	trans-Carveyl acetate	-	-	0.34	-	0.09
33	Nerol acetate	0.56	2.28	9.03	2.13	3.50
34	Geranyl acetate	-	-	0.89	-	0.22
35	Germacrene D	-	-	0.55	-	0.14
36	γ-Elemene	-	-	1.29	-	0.32
37	Spathulenol	0.3	-	0.91	-	0.30
38	Globulol	-	-	0.26	-	0.07
	Monoterpene hydrocarbons	8.99	10.45	13.62	12.69	11.44
	Oxygenated monoterpenes	85.35	81.46	60.58	77.89	76.32
	Sesquiterpene hydrocarbons	0.00	-	0.55	0.96	0.38
	Oxygenated sesquiterpenes	0.3	-	1.17	-	0.37
	Others	3.74	4.79	16.61	3.59	7.18
	Total identified	98.38	96.7	92.53	95.13	95.69
	Oil yield (%, V/W)	0.31	0.65	0.84	0.75	0.64

-: Not detected.

**Table 2 plants-10-00495-t002:** IC_50_ (mg/mL) values of *S. kaschgaricum* volatile oils and their major constituents on root and shoot length of *A. retroflexus* and *P. annua*.

Test Plants	Organs and Major Constituents	Regression Equation	R^2^	IC_50_
*A. retroflexus* root	stem	y = −11.508x^2^ + 93.403x − 82.097	0.9882	1.824
	leaf	y = −9.5325x^2^ + 79.559x − 60.805	0.9848	1.767
	flower	y = −11.779x^2^ + 93.393x − 76.567	0.9789	1.735
	flowering shoot	y = −5.8052x^2^ + 61.363x − 56.398	0.8291	2.186
	eucalyptol	y = 9.3438x − 18.531	0.9327	6.264
	camphor	y = 5.9152x^2^ − 22.147x + 20.625	0.8819	4.782
	mixture	y = 8.4375x − 7.9375	0.9196	6.867
*A. retroflexus* shoot	stem	y = −9.4749x^2^ + 80.749x − 67.239	0.969	1.856
	leaf	y = −8.705x^2^ + 74.203x − 54.044	0.9746	1.769
	flower	y = −9.65x^2^ + 82.18x − 70.342	0.9835	1.879
	flowering shoot	y = −5.2372x^2^ + 59.054x − 58.134	0.8253	2.300
	eucalyptol	y = 2.6388x^2^ − 9.2084x − 5.0955	0.8247	6.636
	camphor	y = 0.8189x^2^ − 2.4932x + 10.955	0.9145	8.593
	mixture	y = 7.1338x − 27.898	0.9718	10.920
*P. annua* root	stem	y = −7.9365x^2^ + 69.349x − 47.888	0.9572	1.770
	leaf	y = −7.1368x^2^ + 61.667x − 30.127	0.9836	1.593
	flower	y = −6.3977x^2^ + 61.779x − 45.954	0.8971	1.945
	flowering shoot	y = 26.641x − 19.025	0.9214	2.591
	eucalyptol	y = 13.236x − 6.6472	0.9108	4.280
	camphor	y = 2.1033x^2^ − 1.6868x + 23.79	0.8723	3.954
	mixture	y = 8.1633x + 19.125	0.9023	4.782
*P. annua* shoot	stem	y = −8.0478x^2^ + 72.872x − 61.388	0.9209	1.947
	leaf	y = −8.732x^2^ + 73.998x − 52.504	0.9859	1.744
	flower	y = −9.6585x^2^ + 85.289x − 84.178	0.9599	2.048
	flowering shoot	y = −6.9597x^2^ + 71.954x − 80.814	0.8953	2.354
	eucalyptol	y = 11.934x − 14.689	0.9772	5.421
	camphor	y = 8.0984x + 13.475	0.9201	4.51
	mixture	y = 9.3115x + 8.0656	0.945	4.504

R^2^: adjusted coefficient of determination, IC_50_: the inhibitory concentration required for 50% inhibition.

## Data Availability

Not applicable.

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
