# Peer review of "Allelopathic Effect of Serphidium kaschgaricum (Krasch.) Poljak. Volatiles on Selected Species"

_plants, 2021, doi:10.3390/plants10030495_

Round 1

Reviewer 1 Report

This article makes an identification and quantification of the volatile oils of Seriphidium kaschgaricum (Krasch.) Poljak., and then performs an assay with plants extracts on two plants, a dicot and a monocot from the same ecological niche, next to another with the main components in pure form.

Although not very exciting texts, similar articles are regularly published by the involved journal, Plants, and they provide knowledge of potential utility, in this case of possible potential ecological relevance.

In order to improve the manuscript to my eyes, I would like to make the following suggestions, some of them of subjective nature.

First comes the genus of the object plant. Seriphidium instead of Artemisia, although still debatable, I believe the second is the proper taxonomic form (see Hussain, A. (2020). The Genus Artemisia (Asteraceae): A Review on its Ethnomedicinal Prominence and Taxonomy with Emphasis on Foliar Anatomy, Morphology, and Molecular Phylogeny. Proceedings of the Pakistan Academy of Sciences: B. Life and Environmental Sciences, 57(1), 1–28.). As a minimum, the reader shall be informed that a controversy exists. I believe also more interested readers will be found if the name Artemisia is used.

In the intro, with mentions of other similar studies specially from the genus Artemisia/Seriphidium I would have liked a little search on the victim plants, Amaranthus retroflexus and Poa annua, as I believe they exist at least for the second one. There is a short paragraph on the ecological significance of the Seriphidium/Artemisia genus, but I miss something on this concrete species, if it exists. The possible utility of the article hangs on the relevance of the plant. About this plant I have found several data on internet included a putative common name, Xinkiang Silk Artemisia, and some articles on Chinese and Russian I cannot read (Specially one by a Goryaev 1962, on volatile).

The discussion seems for the three middle paragraphs an extension of the introduction. Possibly a division can be established, the data belonging to the Seriphidium/Artemisia genus belong in the discussion and all the other in the introduction.

In the methodology I miss from how many samples was the GC/MS performed, if it was from a single plant or a mixture of plants. I suppose it was just made once for each tissue.

About the conclusions. This is just a descriptive article, no new hypothesis can be found in it. It could have been different if the synergistic actions or the minor components responsible for increasing the allelopathic effect would be identified. Maybe a next time.

Reviewer 2 Report

In this study, the authors investigated the allelopathic effect of S. kaschgaricum by determining the inhibitory effects of VOCs produced from S. kaschgaricum on the other two receiver plants, and the main products in the oil generated from S. kaschgaricum were identified. Overall, the authors made a clear description of the experimental results. However, neither the main significance of this study was well stated, nor some of the experimental techniques/steps were adequately provided (Please see comments below). Furthermore, some minor issues might be corrected to improve the quality of the whole manuscript. Thus, the reviewer will not suggest this manuscript being published in Plants unless all the concerns are well addressed.   

Major concerns:

  1. In the Abstract, the authors only stated the experimental results and mentioned this is the first study in related field. However, this reviewer failed to see the significance of the whole study, at least couldn’t find it in the Abstract, making it difficult to benefit for other readers and scientists to acknowledge/use/expand research from this study. Please consider adding the conclusion and significance in the Abstract rather than simply describing results.
  2. Another major concern is, the allelopathic effects of VOCs and the phytotoxic effects of the volatile oils are not well connected. These were two separate experiments, VOCs are products automatically released, but the volatile oils were generated after hydro-distillation. Even though the components and phytotoxicity of plant oils were characterized, it didn’t help explain the phytotoxicity induced by VOCs. Alternatively, at least, more discussion should be involved between these two parts, in terms of potential correlations.  

Minor issues:

  1. In Figure 1, please revise the font of x-axis titles, and also how are those IC50 values derived from the bar graph? Please provide details in the materials & methods section.
  2. Line 99-100, please change the unit of IC50 values into g/1.5L.
  3. In Table 1, it is believed these components were determined in multiple replicates, please provide standard deviation for these results. Does “-“ mean under LOQ or LOD? And in the Compounds column, what is the “Total” about? Please provide all these details in the table title.
  4. Figure 2 is not very informative, it would be easier to show the data with a regular bar graph like figure 1, with error bars and statistical analysis.
  5. Please delete figure 3 since it only shows the structures of those two compounds, but no further useful information can be extracted.
  6. Please provide the rationale of setting the volatile oils exposure concentrations and the choice of tested receiver plants in materials and methods section.
  7. Line 153, it should be figure 4.
  8. Figure 4, please revise the x-axis, it may not be necessary to repeat showing the unit.
  9. Also, please mention how those concentration-response curves were made, the info about the software etc.
  10. Line 189 “IC50”.
  11. In Table 2, please indicate the unit of these IC50
  12. In section 4.3, please provide more details about the hydro-distillation process, such as the processing time, how many samples were prepared at one time. And also provide any references if necessary.
  13. Line 308, “thickness”.
  14. Line 313, please define “NIST”.
  15. In section 4.4, please provide details about how to prepare the samples before injecting into instrument. And the LOQ and LOD of these compounds measured in this study.

Reviewer 3 Report

Dear authors,

Undoubtedly, the characterization of the composition of the VOC of S. kaschgaricum is most important in this manuscript. The authors attempted to determine the influence of the two main compounds in the mixture (eucalyptol and camphor) on the growth of other species' seedlings. The allelopathic effect of these substances is known, and the presented work does not bring anything new to the understanding of its mechanism; it only states the presence of these compounds in plants of the species under study. Moreover, it turned out that it was another component of the extract responsible for the strong allelopathic effect.

The work is purely descriptive; there are no mechanistic hypotheses and no attempts to explain the observed phenomenon.

Here are some detailed questions and issues to discuss:

  • What are the natural VOC contents in the soil near S. kaschgaricum? Has anyone checked it yet? Or maybe the authors can examine the VOC content range in the soil?
  • Results: Only in Figure 1 the authors marked statistical significance. Without statistics, discussing the other Figures is pointless because we cannot say whether the observed changes in the growth parameters are significant. Please add statistical analysis to the other Figures.
  • Results: Reporting the percentage changes is convenient, but it is not very meaningful without the statistics. The authors should first define the observed changes' significance and only then describe them. Reading the description of the results does not make sense, because for sure, a lot of the differences are irrelevant.
  • Table 1. What is the unit for all columns? Percentages? Please specify in Table.
  • Figure 2. is unclear. I propose to change the Figure to a bar chart. If the Figure 2 duplicates the results from the Table 1, delete the Figure.
  • Figure 3. Understanding the chemical structure of these two molecules is not the authors' work. Be sure to remove it from the manuscript.
  • Figures 4. and 5. It would be best if you changed the chart type. Currently, the line connecting individual points suggests a change in time and work on the same plants, while the variants are independent. Moreover, please start the Y-axis from 0 because the authors did not observe a minus length. It is also necessary to add statistical significance.
  • M&M: What do the authors mean by "whole plant"? Are these shoots with roots? Or only flowering shoots? As far as the authors are concerned about the shoots, I would prefer them to clarify it throughout the manuscript because "whole plant" is extremely misleading.
  • The manuscript is written in good English, though mistakes and typos do occur (even in the word camphor - see line 171). Please check and correct the text carefully.

Reviewer 4 Report

Dear Author,

Your paper entitled “Allelopathic Effect of Seriphidium kaschgaricum (Krasch.) Poljak. Volatiles” addresses a topic of great interest for the readership of Plants. Allelopathic phenomena need to be elucidated as allelopathy can be a driver of plant community structure and ecosystem functioning. 

The research is relevant, and well planned. Anyway, the paper has some flaws that need to be improved mainly regarding the doubtful statistical analysis, which, coupled with an uncertain and not exhaustive discussion makes results not sufficiently far-reaching.

Results are interesting but one main concern of mine is about the endorsement of results. Data in tables and figures cannot be identified as significantly different. This follows an unsuitable description of ANOVA. Also, methods should be improved giving additional details and explanations.

Major revisions are needed before acceptance and specific suggestions are given.

Sincerely

General points

The English would really require some major polishing by a native speaker as many mistakes are spread through the manuscript. Only as examples: accordance sub/verb line 34, lines 79-80, lines 55-56.

Are found - line 52.

Which - is too much used and sometimes incorrectly (only eg line 34, 40, 336).

Some typos are spread throughout the manuscript.

Capital V in Volatiles (title)

The same for Growing in abstract

Line 144 , 234 and many others…

Citations in the text are not always as requested by the Journal and the same is true for the listed references.

Specific points

Title: I would change e.g Allelopathic effects of volatile organic compounds of Seriphidium…. On selected species. Thus, it would more accurately and consistently reflect the major point of the paper, because Allelopathy is species-specific.

Introduction is not well centered and is jumping up and down. Some references have to be included (Please see below) to detail literature findings.

M&M:

Why were plants harvested at the flowering stage? Line 280.

It is not clear if controls were included. If not, as it seems, it should be discussed.

Anova is not properly described. How were assumption verified? And treatments arranged? Were there compound, concentration and organ treatments? From some tables it seems so, thus please better detail.

Results: no statistic details are reported in some figures and tables (eg table 2) thus it is not possible to identify significantly different values. For the others, please add the meaning of the bars in graphs.

You have interesting results that deserve a more suitable statistical endorsement.

Table and figures should be amended. For example, table 2 the first column should be Major compounds and organs (see above about statistical analysis).

Discussion:

Line 206 yield of what?

As a matter of fact, the allelopathic effects of allelochemicals are selective, species specific and concentration dependent. This is a topic you should address in discussion.

Please see:
Ercoli et al. 2007. Allelopathic effects of rye, brown mustard and hairy vetch on redroot pigweed, common lambsquarter and knotweed. Allel. J., 19:249-256.

  1. Santonja, A. Bousquet‐Mélou, S. Greff, E. Ormeño, C. Fernandez. Allelopathic effects of volatile organic compounds released from Pinus halepensis needles and roots. Ecol. Evol., 9 (2019), pp. 8201-8213

Viros J, Fernandez C, Wortham H, Gavinet J, Lecareux C, Ormeño E, Litter of mediterranean species as a source of volatile organic compounds, Atmospheric Environment.

These references could also be useful in the introduction to explain the main reason of the present research.

Lines 270: lower effects on shoot growth compared to root growth on Amaranthus were also reported by Ercoli et al. 2007 and were identified as allelopathic effects due to concentration effects. This reference should be cited here to support your results as you didn’t survey concentrations.

Please re write lines 248-250.

References: some important items are missed. Please  see above.

Round 2

Reviewer 2 Report

All the concerns have been well addressed.

Author Response

All authors greatly appreciate your assistance in improving the quality of our manuscript; we feel we have learned a lot from your valuable comments.

Reviewer 3 Report

Dear authors,

Thank you for your comprehensive answers and for taking the matter seriously. I especially appreciate the amount of data work and the preparation of new charts. The authors also significantly improved the language of the work.

Unfortunately, the work I received is illegible. There are four font colors, and some text is crossed out, some underlined, some graphs are green, some are duplicated. The authors write that they removed the graphs, but they are still present in the revised manuscript. Chaos. I am asking the authors to prepare an up-to-date version of the manuscript containing only up-to-date charts, and any changes in the text compared to the original may be marked with one different font color.

By the way, I suggest that authors consider the importance of information, which is essential for the authors in a given chart. For example, new Figure 2. Is the impact of a given VOC (e.g., Flower) on growth important? Is it possible to compare a given concentration of different VOCs with each other? Please adjust the statistics under this account. Also, extend the Figures' descriptions so that they are self-explanatory, i.e., looking at the chart and reading its legend, I know what has been tested and what I see on the chart without reading the paper.

Reviewer 4 Report

Dear Author,

I would like to applaud your work in amending the manuscript.

The revised paper has been greatly upgraded and all responses to my observations are satisfying.

Introduction is now adequate, methods have been properly described, results and discussion are well organized.

The paper can now be accepted.

Congratulations

Author Response

(The authors gave the same response as above.)
